# Long-Term Ingestion of Sicilian Black Bee Chestnut Honey and/or D-Limonene Counteracts Brain Damage Induced by High Fat-Diet in Obese Mice

**DOI:** 10.3390/ijms24043467

**Published:** 2023-02-09

**Authors:** Simona Terzo, Pasquale Calvi, Domenico Nuzzo, Pasquale Picone, Mario Allegra, Flavia Mulè, Antonella Amato

**Affiliations:** 1Dipartimento di Scienze e Tecnologie Biologiche, Chimiche e Farmaceutiche (STEBICEF), Università degli Studi di Palermo, 90128 Palermo, Italy; 2Dipartimento di Biomedicina, Neuroscienze e Diagnostica Avanzata (Bi.N.D.), Università degli Studi di Palermo, 90127 Palermo, Italy; 3National Research Council of Italy—CNR, Institute for Biomedical Research and Innovation—IRIB, 90146 Palermo, Italy

**Keywords:** obesity, high-fat diet, neuronal damage, neurodegeneration, honey, D-limonene

## Abstract

Obesity is linked to neurodegeneration, which is mainly caused by inflammation and oxidative stress. We analyzed whether the long-term intake of honey and/or D-limonene, which are known for their antioxidant and anti-inflammatory actions, when ingested separately or in combination, can counteract the neurodegeneration occurring in high fat diet (HFD)-induced obesity. After 10 weeks of HFD, mice were divided into: HFD-, HFD + honey (HFD-H)-, HFD + D-limonene (HFD-L)-, HFD + honey + D-limonene (HFD-H + L)-fed groups, for another 10 weeks. Another group was fed a standard diet (STD). We analyzed the brain neurodegeneration, inflammation, oxidative stress, and gene expression of Alzheimer’s disease (AD) markers. The HFD animals showed higher neuronal apoptosis, upregulation of pro-apoptotic genes Fas-L, Bim P27 and downregulation of anti-apoptotic factors BDNF and BCL2; increased gene expression of the pro-inflammatory IL-1β, IL-6 and TNF-α and elevated oxidative stress markers COX-2, iNOS, ROS and nitrite. The honey and D-limonene intake counteracted these alterations; however, they did so in a stronger manner when in combination. Genes involved in amyloid plaque processing (APP and TAU), synaptic function (Ache) and AD-related hyperphosphorylation were higher in HFD brains, and significantly downregulated in HFD-H, HFD-L and HFD-H + L. These results suggest that honey and limonene ingestion counteract obesity-related neurodegeneration and that joint consumption is more efficacious than a single administration.

## 1. Introduction

Neurodegenerative diseases (NDs), including Alzheimer’s disease (AD), are characterized by the progressive loss of neurons in areas of the brain, leading to cognitive and functional deterioration. NDs represent a serious problem because they affect about 50 million patients worldwide and this number is estimated to reach 115 million in 2050 [1]. Different factors contribute to the onset and progression of neurodegeneration such as aging, genetics, environment [2] and oxidative stress and inflammation [3,4]. Moreover, obesity and diabetes increase the risk of developing dementia and AD. In fact, the neurodegenerative process is exacerbated by obesity or diabetes, leading to the concept of metabolism-dependent neurodegeneration [5]. Indeed, insulin receptor down-regulation has been observed in the brains of patients with AD [6] confirming the theory that AD may be considered as “type 3 diabetes” [7]. Furthermore, studies on animal models pointed out that obesity affects learning and memory [8,9] and long-term ingestion of high-fat diet (HFD) in rodents is responsible for neuronal loss and synaptic plasticity damage [10,11,12,13].

NDs are as yet incurable and strongly debilitating for the patients. Nevertheless, current research is pushing towards effective therapies [14,15]. Numerous nutraceuticals and/or functional foods are considered protective and/or therapeutic against the metabolic dysfunctions and the related neurodegeneration [16,17,18,19,20]. For example, foods rich in antioxidants, micronutrients, phytochemicals, essential oils, and probiotics have been found to be helpful in maintaining body weight and reducing the incidence of neurodegenerative diseases [1,20].

In particular, honey might prove useful in the treatment of chronic diseases linked to oxidative stress and inflammation due to its high content in polyphenols [21]. Although the composition of honey is variable depending on various factors such as botanical origin, geographical region and climatic conditions, most of the polyphenols present in honey are flavonoids and phenolic acid derivatives that possess anti-inflammatory and neuroprotective properties [22].

More specifically, our recent investigation demonstrated the ability of honey consumption to prevent HFD-dependent neuronal injury. In particular, a 16-week-intake of Sicilian black bee chestnut honey, which is particularly rich in kaempferol and quercetin [22], prevented peripheral and central insulin resistance and neuroinflammation in mice fed with a hyperlipidic diet [23]. This neuroprotective effect proved to be mainly due to the positive modulation of brain genes involved in insulin signaling, neuroinflammation and apoptosis [23]. However, it remains to be investigated whether the long-term ingestion of honey is able to revert obesity-related metabolic dysfunctions and related neurodegeneration.

Recently, D-limonene (1-methyl-4-(1-methylethenyl) cyclohexane), a monocyclic monoterpene that is the major constituent of citrus essential oils, has also received notable scientific interest due to its ability to mitigate inflammation and oxidative stress and reduce apoptotic cell death [24]. In fact, it possesses antidiabetic, antioxidant, anti-inflammatory, antinociceptive and anticancer properties [25]. In animal models, D-limonene has been reported to alleviate obesity-related metabolic disorders [26,27]. However, although D-limonene has recently been shown in vitro to inhibit acetylcholinesterase [28] and to exert beneficial effects in the *Drosophila* AD model by reducing oxidative stress and neuroinflammation [29], data on neuroprotective actions against neuronal damage caused by HFD are lacking. Moreover, a recent study suggested that the usage of D-limonene together with other drugs, such as aminoguanidine, is more efficient in the prevention of secondary complications in diabetes in comparison to single treatment [30].

Therefore, the present research was undertaken with the purpose of exploring whether honey, administered alone or in combination with D-limonene, can represent a potential dietary supplement that can aid in ameliorating or reverting HFD-caused brain damage. In this view, we investigated the effects of the long-term ingestion of honey and D-limonene, separately or in combination, on brain damage in HFD mice when the pathological conditions were overt.

## 2. Results

### 2.1. Body Weight, Glycaemia and Serum Lipids

As shown in Figure 1A, at the end of the experimental protocol, HFD mice were significantly heavier than STD mice. The weight gain of HFD-L and HFD-H + L mice was significantly lower than that of HFD animals. The fasting blood glucose concentration of HFD mice was significantly higher than that of the STD group. HFD-H and HFD-H + L mice had similar fasting blood glucose concentrations to the HFD group. However, the D-limonene supplementation markedly reduced the fasting blood glucose levels induced by HFD (Figure 1B). The lipid profile of mice that were fed with the different diets is represented in Figure 1C. Total cholesterol and triglyceride levels, that were high in the plasma of the HFD mice compared to STD group, did not significantly differ in HFD-H, HFD-L and HFD-H + L mice, suggesting that honey and D-limonene, alone or in combination, did not impact on the lipid metabolism of obese mice.

### 2.2. Neurodegeneration: TUNEL Assay

Neurodegeneration has been suggested to be associated with cell apoptosis. To identify whether apoptotic cells were present in the brain tissues of the different groups of mice, we used the TUNEL assay. A higher number of TUNEL-positive cells was observed in the cortex of HFD mice in comparison with STD mice. As shown in Figure 2, neuronal apoptosis resulting from a high-fat diet was significantly decreased in the cortex of HFD-H, HFD-L, and HFD-H + L mice, suggesting that both honey and D-limonene contributed to neuroprotective effects. Interestingly, the diet containing honey and D-limonene together was more efficacious than the single supplement.

### 2.3. Pro-Apoptosis and Anti-Apoptosis Genes Expression

In this work, the gene expression of the most important regulators of apoptosis. The pro-apoptotic factors *FAS-L*, *P27*, and *BIM* were significantly upregulated in mouse brain tissues from the HFD group compared to the STD group. A high-fat diet supplemented with honey, D-limonene or honey plus D-limonene significantly decreased the gene expression levels of all investigated factors, suggesting a reduced presence of neurons that undergo programmed cell death (Figure 3A,B). On the contrary, the brain gene expression of factors that help neuronal survival, such as BDNF and BCL2, was decreased in the HFD mice compared to the STD group. This down-regulation induced by HFD was counteracted by the simultaneous ingestion of honey or D-limonene. HFD-H + L proved to be the most efficacious diet to increase BCL2 and BDNF expression (Figure 3C,D).

### 2.4. Brain Pro-Inflammatory Gene and Protein Expression

To determine whether honey and D-limonene, together or separately, reduced neuroinflammation, we examined the brain expression of some pro-inflammatory cytokines and other proteins, which are markers of inflammation. The IL-1β, IL-6 e TNF-α increased expression, found in HFD brains, was reduced by honey or D-limonene ingested separately, and it returned to control levels in the brain of HFD-H + L mice, suggesting that the combined administration of honey and D-limonene was more efficacious than single administration (Figure 4A,B). Moreover, the elevated expression of COX-2 and iNOS induced by HFD, was mitigated by honey, D-limonene, and honey plus D-limonene (Figure 5A,B).

### 2.5. Brain Oxidative Stress

Increasingly, studies have demonstrated that oxidative stress is critical for neuronal injury. Therefore, we determined the effect of the different supplemented diets on ROS generation and nitrite content in the brain of the different groups of animals. After a 20-week HFD administration, ROS generation assessed with H_2_DCF-DA was significantly increased in HFD brain compared not only to STD, but also to HFD-H, HFD-L, and HFD-H + L (Figure 6A). Moreover, we found a significant increase in nitrite levels in the brains of HFD obese animals in comparison with STD animals. HFD-H, HFD-L, and HFD-H + L mice showed nitrite values that were significantly lower than those of HFD mice (Figure 6B).

### 2.6. Expression of Genes Involved in AD

Using a Mouse Alzheimer’s Disease RT^2^ Profiler PCR Array we analyzed expression changes of genes involved in the onset, development and progression of Alzheimer’s disease in the different groups of animals. Among them, there are genes that contribute to amyloid beta-peptide (Aβ) generation, clearance and degradation but also genes related to neuronal toxicity. The list of genes is shown in Appendix A. We focused on the gene expression levels that were affected more than two-fold among the analyzed groups. The results showed that in the HFD brains, various genes involved in the processing of Amiloid β Precusor (APP) and TAU (*Aplp1*, *Aplp2*, *App*, *Apba3*, *Apbb2*, *Apoe*, *Ckk5*, *Clu*, *Ctsl*, *Mapt*, *Prkca*, *Prkce* and *Hsd17b10*), in synaptic function (*Ache*), in AD-related iperphosphorylation (*Gsk3α*, *GCdk5* and *Prkca*), and in inflammation (*MPO* and *Il-1α*) (Table 1) were upregulated in comparison with lean brains. These abnormal expressions were significantly ameliorated in the brain of obese animals fed with honey, D-limonene and honey plus D-limonene with a major improvement in the HFD-H group (Table 1).

## 3. Discussion

The results of the present study suggest that long-term intake of Sicilian black bee chestnut honey and/or D-limonene, ingested separately or in combination, can protect central neurons against HFD-induced cerebral damage by reducing oxidative stress and neuroinflammation. To our knowledge, our study is the first report on the neuroprotective effects of D-limonene against the damage induced by HFD.

Epidemiological human studies pointed out that a high-calorie diet is associated with worse performance on cognitive tasks [31]. It increases the risk of dementia because high lipid content causes oxidative stress and neuronal dysfunctions [32]. Indeed, high stress oxidative triggers the up-regulation of pro-inflammatory factors leading to neuroinflammation [33]. However, different biological mechanisms including insulin resistance, developmental disturbances, altered membrane functioning, and altered vascularization have been involved in HFD-induced neuronal damage and cognitive decline [5,32].

In our experiments we used mice which, following chronic consumption of HFD, developed obesity accompanied by hyperglycemia, dyslipidaemia, insulin resistance [34,35,36], activation of amyloidogenic pathways, neuroinflammation and neurodegeneration [10,18,37,38,39]; consequently, they are suitable for verifying the potential effects of functional food/phytochemicals on neuronal survival. First of all, we analyzed the presence of apoptosis in the cerebral cortex and the gene expression of pro- and anti-apoptotic factors in the brains of the different animal groups. It is well known that apoptosis plays a key role in the pathogenesis of neurodegenerative diseases [40], involving mainly the BCL-2 protein family. This family includes proteins that control the mithocondrion membrane permeability such as Bax, Bim (pro-apoptotic proteins) and BCL-2, Bcl-xL, Bcl-w (anti-apoptotic proteins). Additionally, FAS ligand (FAS-L) has been involved in neuronal death [41] and P-27, an inhibitor of cyclin-dependent kinase, has been reported to promote neuronal apoptosis induced by the neurotoxic αβ42 peptide [42]. According to our previous reports [10,17,18], our results confirmed the presence of neurodegeneration caused by HFD as suggested by the increase of apoptotic neurons in the brain cortex of obese mice in comparison with STD mice. In HFD-H or HFD-L cerebral cortexes, the level of apoptotic neurons was significantly reduced suggesting that the daily ingestion of honey or D-limonene inhibits programmed cellular death. Moreover, honey and D-limonene ingested in combination further decreased the apoptotic neuron number, suggesting a synergistic neuroprotective action. The results from molecular analysis also supported our hypothesis on the neuroprotective effect of honey and D-limonene. In fact, the pro-apoptotic gene up-regulation and the anti-apoptotic gene down-regulation that was found in the HFD brain was attenuated in HFD-H, HFD-L and HFD-H + L animal groups. We also found a down-regulation of BDNF in the HFD brain, which was in accordance with previous studies that reported reductions in levels of BDNF in the hippocampus of obese rodents [11,43] as a consequence of increased oxidative stress [44]. However, honey and D-limonene when separately ingested increased the BDNF gene expression; even more so when ingested in combination, suggesting that an increase of survival factors can also be responsible for the observed beneficial effects.

Neurodegeneration can be triggered by various pro-inflammatory and neurotoxic mediators, such as IL-1β, IL-6, and TNF-α, and neuroinflammation is strictly associated with oxidative stress [45]. Indeed, several studies demonstrated that neuroinflammation is linked to high levels of ROS and high expression of AD biomarkers in the brains of HFD mice [10,46,47]. Because both honey and D-limonene have been reported to possess anti-inflammatory and antioxidant properties, leading to the assumption that they could be used as a supplement in anti-inflammatory therapies [21,48,49] we examined and compared the expression of pro-inflammatory factors, the levels of oxidative stress and nitrite in the brains of the different animal groups. The results suggested that HFD increases the gene expression of inflammatory cytokines (IL-1β, IL-6, TNF-α) and other proteins, markers of inflammation (i-NOS and COX-2) and ROS and nitrite levels in the brain as previously shown [10,18,32,50,51]. Interestingly, long-term ingestion of honey or D-limonene, and even more so, the combined ingestion of honey and D-limonene reduced the inflammatory and oxidative stress markers suggesting once more a beneficial action against damage induced by HFD in the brain. We can only speculate about the honey compounds responsible for the observed beneficial effects, which generally have been attributed to polyphenols [21]. However, it is noteworthy that we used Sicilian black bee chestnut honey, whose kaempferol and quercetin levels corresponded to 69% of the total content [22]. Quercetin as well as kaempferol can cross the blood–brain barrier [52]. Quercetin has been reported to protect neurons from oxidative stress and inflammation and to have beneficial properties against mechanisms involved in AD in different in vitro and in vivo models [53], and kaempferol can act positively in various models of neurodegenerative diseases [54,55].

Although recent research using a Drosophila AD model suggested that D-limonene has a neuroprotective action against Aβ_42_-induced toxicity associated with its antioxidant and anti-inflammatory properties [29], the effects of D-limonene on AD have not been well-studied yet. Therefore, by using a mouse Alzheimer’s disease microarray, we have analyzed and compared the expression of genes involved in amyloid beta-peptide (Aβ) generation and processing and/or genes related to neuronal toxicity in the brains of different mouse groups. The results clearly suggest that long-term HFD feeding promotes the expression of genes associated with AD, including *Ache*, *App*, *Apba3*, *Apbb2*, *Aplp1*, *Aplp2*, *Apoe*, *CdK5*, *Clu*, *Ctls*, *GSK3α*, *Hsd17b10*, *Mapt*, *Psen1*, *Prkca*,*Prkcb* and genes linked to inflammation such as *Mpo and Il1α* [56,57]. However, these deleterious changes in gene expression were counteracted in the brains of HFD-H, HFD-L and HFD-H + L, suggesting that the increased neurotoxicity induced by HFD may be mitigated by long-term ingestion of honey and D-limonene, both separately and in combination. In particular, the down regulation of *App*, *Apba3*, *Apbb2*, *Aplp1*, *Aplp2*, *Apoe* and *Psen1* could suggest that the eventual endogenous APP generation and processing were reduced after the long-term ingestion of honey and D-limonene [58]. Moreover, *Cdk5*, a promoter of neuronal death [59] and *Clu*, encoding clusterin, a protein involved in several processes such as suppression of the complement system, lipid transport, and neuronal cell death and cell-survival mechanisms, whose levels are increased in AD [60], were mitigated by the intake of honey and D-limonene either alone or in combination.

## 4. Materials and Methods

### 4.1. Animals and Diets

Male C57BL/6 mice, purchased from Envigo (S.Pietro al Natisone, Udine, Italy) were maintained in the ATeN center animal house according to the European guide lines. The animals (4-weeks old) were housed (2 mice/cage) in a temperature- (23 ± 1 °C) and relative humidity (55% ± 5%)-controlled facility, under a 12-h light–dark cycle, according to the Italian legislative decree n. 26/2014 and were approved by the Ministry of Health (Rome, Italy; Authorization n. 891/2018-PR).

After two weeks of acclimatization, 8 mice were fed a standard diet (STD) (negative control) containing protein 20.0%, fat 10.0%, carbohydrate 70.0%, *w*/*w*, and water (code 4RF25, Mucedola, Milan, Italy), and 32 mice were fed a HFD, containing protein 20.0%, fat 60.0%, carbohydrate 20.0%, *w*/*w* (PF4215, Mucedola, Milan, Italy) for 10 weeks to induce obesity. Subsequently, HFD mice were divided randomly into four groups Then, four groups (n = 8/group) were created from the HFD mice: one group received HFD, the second group received HFD supplemented with honey (45 mg per day/mouse) (HFD-H), the third received HFD supplemented with D-limonene (0.5% *w*/*w*) (HFD-L) and the last one received HFD supplemented with honey and D-limonene in combination at the same doses (HFD-H + L), for another 10 weeks. The doses of D-limonene (Sigma—St. Louis, MO, USA) and honey (Prezzemolo and Vitale Supermarket, Palermo, Italy) were taken from the literature [23,27,61] and added to the HFD cow in a percentage amount that was useful so as not to change the HFD caloric value. Body weight and food intake were monitored every week.

At the end of the experimental protocol (20th week), biochemical analyses were performed on blood collected from the tail vein and then the animals were sacrificed. The aorta was perfused with a buffer solution of Dulbecco and the right atrium was incised to allow outflow. Brains were rapidly explanted, weighed and coronally cut into two halves. One part was fixed in 4% formalin was utilized for histological investigation; the other half was ice-covered and used for molecular analysis.

### 4.2. Biochemical Analyses

Glucose concentration was measured using a glucometer (GlucoMen LX meter, Menarini, Florence, Italy) in overnight fasting mice. Plasma total cholesterol and triglyceride concentrations were determined using the ILAB 600 Analyzer (Instrumentation Laboratory, Bedford, MA, USA).

### 4.3. Apoptosis Investigation

The Tunel assay was used to determine the level of apoptosis (Promega, Madison, WI, USA) in the cerebral cortex sections, following the manufacturer’s instructions. The values of the damaged nuclei were counted by two blind investigators and the ratio of apoptotic nuclei in respect of normal nuclei was calculated.

### 4.4. Reactive Oxygen Species Analysis

To determine the reactive oxygen species (ROS), 5 mg of brain tissue was homogenized with 1 mL of cold PBS1X and 10 μL of protease inhibitors (Amersham Life Science, Munich, Germany). The preparate brain homogenates were incubated with 1 mM dichlorofluorescein diacetate (DCFH-DA) at room temperature in the dark for 15 min, then the fluorescence was measured by fluorimeter (GloMax^®^ Plate Reader, Promega, Milano, Italy) with an excitation filter set at 485 nm and an emission filter set at 530 nm. ROS levels were expressed as a percentage of the fluorescence emitted by STD cerebral samples.

### 4.5. Determination of Nitric Oxide (NO) Levels

The level of nitric oxide (NO) in the brains was evaluated by using Griess reagent (Thermo Fisher Scientific Inc., Waltham, MA, USA). Briefly, 5 mg of brain tissue was homogenized with 1 mL of PBS1X and centrifuged at 14,000 rpm, for 30 min at 4 °C. 100 μL of supernatant was incubated with equal volumes of Griess reagent (1% sulphanilamide in 5% phosphoric acid and 0.1% N-(1-naphthyl)-ethylenediamine), the absorbance was immediately read at 520 nm in a microplate reader (GloMax^®^ Plate Reader, Promega).

### 4.6. Molecular Analyses

Whole brain was used to extract RNA by using a RNeasy plus Mini Kit (Qiagen, Valencia, CA, USA). Subsequently, by using High-Capacity cDNA Reverse Transcription Kit (Applied Biosystems, Waltham, MA, USA). cDNA was prepared by 2 ng of total RNA. Then the expression of target genes was performed by using Reverse Transcription Polymerase Chain Reaction (RT-PCR) with the subsequent primers: β-actin *For* 5′-CGGGATCCCCGCCCTAGGCACCAGGGT-3′; *Rev* 5′-GGAATTCGGCTGGGGTGTTGAAGGTCTCAAA-3′; for pro-inflammatory factors: IL-1β *For* 5′-CATGGGATGATGATAACCTGCT-3′; *Rev* 5′-CCCATACTTTAGGAAGACACGATT-3′; IL-6 *For* 5′-CTGGTGACAACCACGGCCTTCCCT-3′; *Rev* 5′-ATGCTTAGGCATAACGCACTAGGT-3′; TNF-α *For* 5′-AGCCCACGTCGTAGCAAACCA-3′; *Rev* 5′-GCAGGGGCTCTTGACGGCAG-3′; for pro-apoptotic factors: FAS-L *For* 5′-CAAGTCCAACTCAAGGTCCATGCC-3′; *Rev* 5′-AGAGAGAGCTCAGATACGTTTGAC-3′; BIM *For* 5′-AACCTTCTGATGTAAGTTCT-3′; *Rev* 5′-GTGATTGCCTTCAGGATTAC-3′; p27 *For* 5′-TGCGAGTGTCTAACGGGAG-3′; *Rev* 5′-GTTTGACGTCTTCTGAGGCC-3′; for anti-apoptosis factors: BCL-2 *For* 5′-ATGTGTGTGGAGAGCGTCAA-3′; *Rev* 5′-AGAGACAGCCAGGAGAAATCA-3′; BDNF *For* 5′-GGCTGACACTTTTGAGCACGTC-3′; *Rev* 5′-CTCCAAAGGCACTTGACTGCTG-3′. The amplification cycles comprised denaturation (45 s at 95 °C), annealing (45 s at 52 °C) and elongation (45 s at 72 °C), for 40 cycles. The amplification products were visualized by ultraviolet light using E-Gel GelCapture (Thermo Fisher Scientific, Monza, Italy) after separation on agarose gel. The quantification of gene expression was obtained by using E-Gel GelQuant Express Analysis Software (version 1.14.6.0 (Dongle)) (Thermo Fisher Scientific, Monza, Italy). The signal intensity of the products was normalized to its respective β-actin signal intensity.

Protein expression. Brains dissected from the experimental animals were homogenized in ice-cold solubilization buffer (50 mM Tris-HCl pH 7.4, 150 mM NaCl, 1 mM DDT, 1% Triton X-100, 24 mM sodium deoxycholate, 0.01% SDS, 10 mM sodium pyrophosphate, 100 mM sodium fluoride, 10 mM sodium orthovanadate, 1.5 µM aprotinin, 1 mM phenylmethanesulfonylfluoride and 2.1 µM leupeptin) and centrifuged at 12,000× *g* at 4 °C for 30 min. Then, the supernatants were used for protein determination, as previously described [62]. Samples containing 50 µg protein were resolved by SDS-PAGE electrophoresis on 12% acrylamide gels and transferred to nitrocellulose membranes. After blocking for 2 h in 5% (*w*/*v*) skimmed dry milk, the membranes were incubated in the presence of primary antibodies overnight at 4 °C (Santa Cruz, Milan, Italy, 1:1000 dilution): anti-COX-2 (sc-376861), anti-iNOS (sc-7271). Subsequently, the samples were incubated with the secondary for 90 min. HRP-conjugated antibodies (Dako, Milan, Italy, 1:10,000 dilution) and chemiluminescent bands were detected by a C-Digit Blot Scanner (LI-COR, Lincoln, NE, USA) and densitometric analysis was used to analyze band intensities, by using LI-COR Image Studio 4.0.

### 4.7. RT^2^ Profiler PCR Array

Mouse Alzheimer’s disease array (Alzheimer’s Disease RT^2^ Profiler PCR Array, QIAGEN, Monza, Italy) was used in order to analyze factors in HFD brains that are involved in the onset, development and progression of AD. The 96 genes reported in the plate are listed in the Appendix A.

RNA from whole brains was utilized. A High-Capacity cDNA Reverse Transcription kit (Applied Biosystems, Bedford, MA, USA) was used to synthetize cDNA from 2 ng of RNA. The array was executed by using a StepOne Real-Time instrument (Applied Biosystem) and the results were obtained through the relative quantification method (2^−ΔΔCT^).

We chose to highlight only the genes showing changes in the expression levels that were more than two-fold among the different groups analyzed (HFD vs. Lean; HFD-H vs. HFD; HFD-L vs. HFD; HFD-H + L vs. HFD).

### 4.8. Statistical Analysis

The results are presented as mean values ± the standard error of the mean SEM. The number of animals is indicated with the letter ‘n’. The comparison between the groups was performed by ANOVA, and then a Bonferroni post hoc test was used. All the analyses were obtained using Prism 6.0, GraphPad software (San Diego, CA, USA). Results with a *p*-value ≤ 0.05 were considered statistically significant.

## 5. Conclusions

In conclusion, our results confirm that HFD causes detrimental effects on AD-related neuropathological and neuroinflammatory pathways leading to neurodegeneration. However, the long-term ingestion of honey and D-limonene, either separately or in combination, is able to counteract and to ameliorate the cerebral stressing conditions related to HFD-induced metabolic disorders.

## Figures and Tables

**Figure 1 ijms-24-03467-f001:**
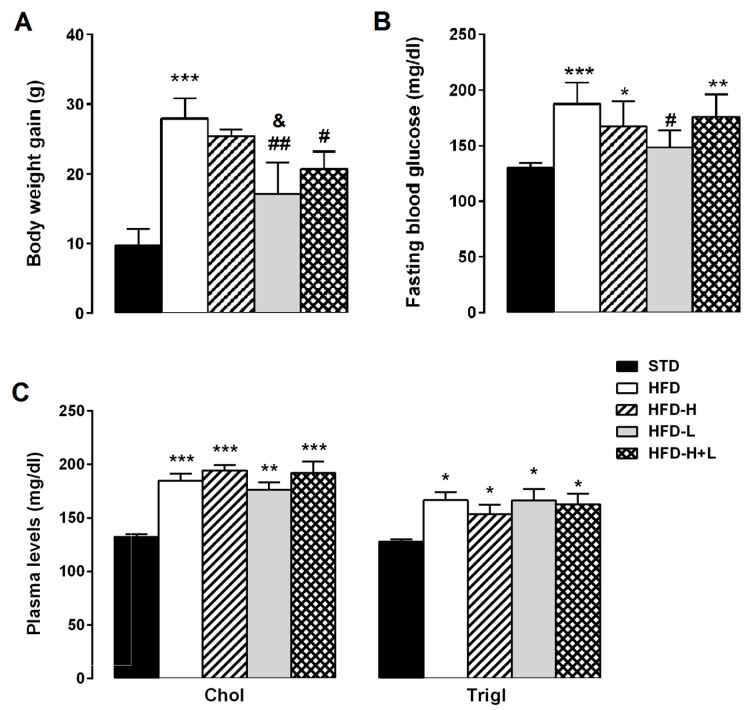
Metabolic parameters. (**A**) body weight gain, (**B**) fasting blood glucose concentrations and (**C**) plasma levels of cholesterol and triglycerides in obese mice. Data are mean values ± S.E.M. (*n* = 8/group). * *p* < 0.05, ** *p* < 0.01, *** *p* < 0.001 vs. STD mice; ^#^ *p* < 0.05, ^##^ *p* < 0.01; ^&^ *p* < 0.05 vs. HFD-H mice.

**Figure 2 ijms-24-03467-f002:**
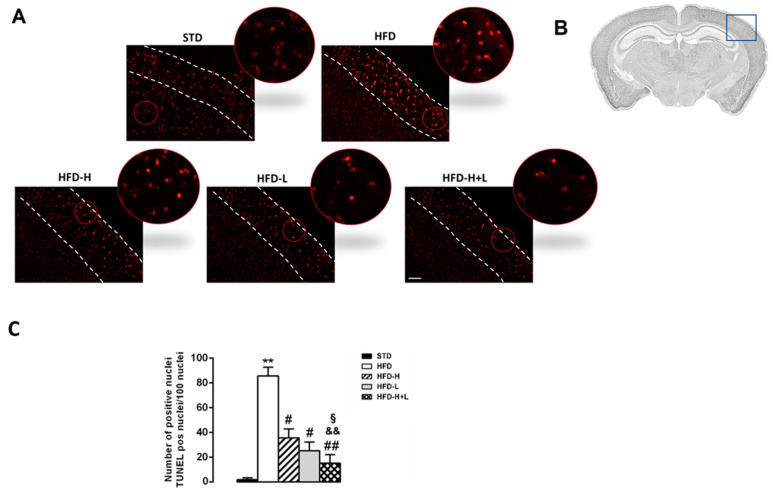
Neurodegeneration. (**A**) TUNEL assay; the outlined area is enlarged in the circle; white dotted lines represent the mouse brain cortex. (**B**) Schematic representation of cerebral cortex of positive areas. (**C**) Number of apoptotic nuclei in cerebral cortex of STD, HFD, HFD-H, HFD-L and HFD-H + L mice. Data are mean values ± S.E.M. (n = 8/group). ** *p* < 0.01 vs. STD mice; ^#^ *p* < 0.05, ^##^ *p* < 0.01 vs. HFD mice; ^&&^ *p* < 0.01 vs. HFD-H mice; ^§^
*p* < 0.05 vs. HFD-L mice. Microscope magnification 10×. Scale bar, 200 μm.

**Figure 3 ijms-24-03467-f003:**
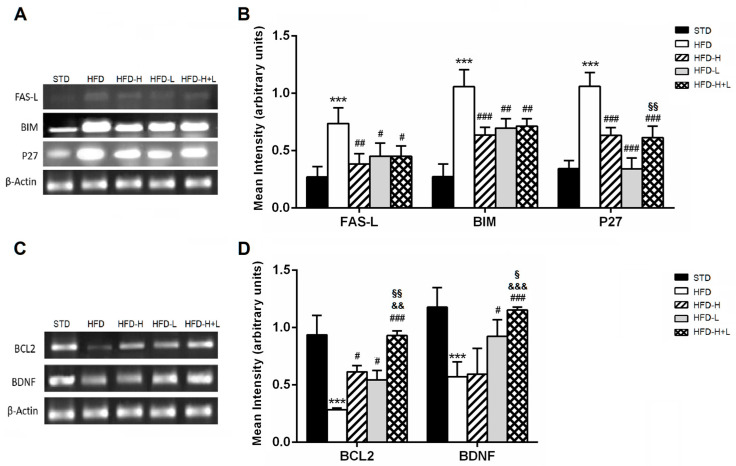
Apoptosis. (**A**) result of the RT-PCR and (**B**) mRNA levels of pro-apoptotic genes: *FAS-L*, *BIM* and *P27* in the mouse brain of different groups; (**C**) Representative image of the RT-PCR results and (**D**) mRNA levels of survival genes: *BCL2* and *BDNF* in the mouse brain of different groups. Data are mean values ± S.E.M. (n = 8/group). *** *p* < 0.001 vs. STD mice; ^#^ *p* < 0.05, ^##^ *p* < 0.01, ^###^ *p* < 0.001 vs. HFD mice; ^&&^ *p* < 0.01, ^&&&^
*p* < 0.001 vs. HFD-H mice; ^§^
*p* < 0.05, ^§§^ *p* < 0.01 vs. HFD-L mice.

**Figure 4 ijms-24-03467-f004:**
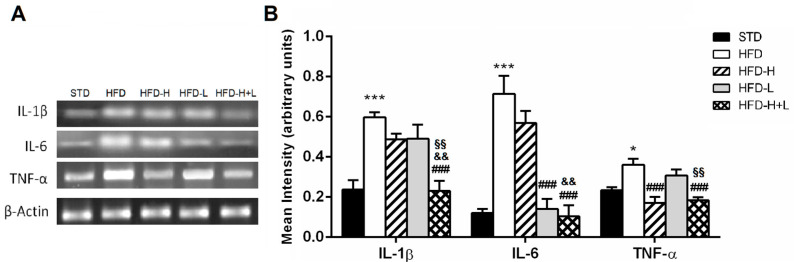
Brain inflammation. (**A**) Representative image of the RT-PCR results and (**B**) mRNA levels of IL-1β, TNF-α and IL-6 in the mouse brain of different groups. Data are mean values ± S.E.M. (n = 8/group). * *p* < 0.05, *** *p* < 0.001 vs. STD mice; ^###^ *p* < 0.001 vs. HFD mice, ^&&^ *p* < 0.01 vs. HFD-H mice; ^§§^ *p* < 0.01 vs. HFD-L mice.

**Figure 5 ijms-24-03467-f005:**
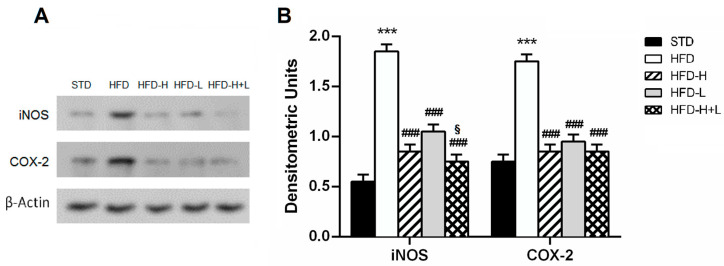
Brain stress. (**A**) protein expression levels of COX-2 and iNOS in the mouse brain of different groups; (**B**) densitometric analysis of iNOS and COX-2 protein levels normalized for β-actin levels. Data are mean values ± S.E.M. (n = 8/group). *** *p* < 0.001 vs. STD mice; ^###^ *p* < 0.001 vs. HFD mice; ^§^
*p* < 0.05 vs. HFD-L mice.

**Figure 6 ijms-24-03467-f006:**
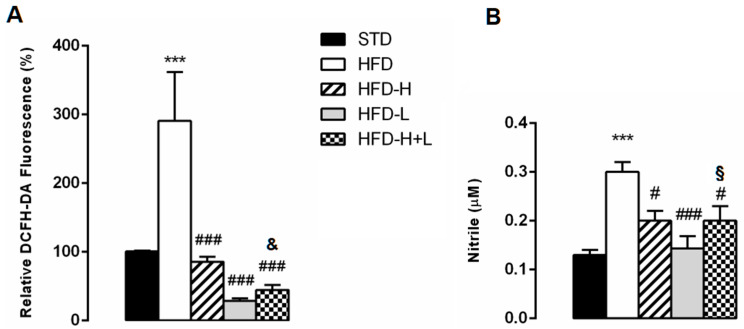
Brain oxidative stress. (**A**) Levels of ROS in the brain of STD, HFD, HFD-H, HFD-L and HFD-H + L mice. (**B**) Nitrite concentration in the brain of STD, HFD, HFD-H, HFD-L and HFD-H + L mice. Data are mean values ± S.E.M. (n = 8/group). *** *p* < 0.001 vs. STD mice; ^#^ *p* < 0.05, ^###^ *p* < 0.001 vs. HFD mice; ^&^ *p* < 0.05 vs. HFD-H mice; ^§^
*p* < 0.05 vs. HFD-L mice.

**Table 1 ijms-24-03467-t001:** Gene expression profiles involved in AD in HFD/Lean, HFD-H/HFD, HFD-L/HFD and HFD-H + L/HFD, which were significantly downregulated by 2-fold.

Gene Name	Protein	HFD/Lean	HFD-H/HFD	HFD-L/HFD	HFD-H + L/HFD
*Ache*	Acetylcholinesterase	2.10	−30.84	−12.01	−16.31
*Apba3*	Amyloid beta (A4) precursor protein-binding, family A, member 3	44.94	−59.17	−30.02	−29.88
*Apbb2*	Amyloid beta (A4) precursor protein-binding, family B, member 2	2.60	−18.46	−10.71	−17.39
*Aplp1*	Amyloid beta (A4) precursor-like protein 1	5.17	−8.79	−8.53	−5.00
*Aplp2*	Amyloid beta (A4) precursor-like protein 2	15.97	−45.55	−15.49	−17.72
*Apoe*	Apolipoprotein E	5.16	−26.20	−9.36	−9.77
*App*	Amyloid beta (A4) precursor protein	6.05	−15.85	−9.94	−10.12
*Bdnf*	Brain derived neurotrophic factor	−17.99	−5.49	−3.44	−2.67
*Cdk5*	Cyclin-dependent kinase 5	2.92	−36.93	−17.68	−13.99
*Clu*	Clusterin	26.50	−4.04	−3.76	−2.19
*Ctsl*	Cathepsin L	5.08	−13.34	−4.65	−4.90
*Gsk3* *a*	Glycogen synthase kinase 3 α	7.20	−22.22	−14.11	−15.35
*Hsd17b10*	Hydroxy steroid deydrogenase10	5.87	−15.93	−9.38	−11.82
*Il1a*	Interleukin 1 α	4.31	−90.55	−49.60	−98.59
*Mapt*	Microtubule-associated protein tau	44.58	−23.78	−9.87	−8.28
*Mpo*	Myeloperoxidase	14.87	−33.10	−14.25	−28.87
*Prkca*	Protein kinase C, alpha	6.17	−16.16	−15.98	−10.33
*Prkce*	Protein kinase C, epsilon	5.89	−16.58	−12.56	−16.61
*Psen1*	Presenilin 1	4.68	−225.85	−152.13	−193.57

## Data Availability

Not applicable.

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
