# Peer review of "Long-Term Ingestion of Sicilian Black Bee Chestnut Honey and/or D-Limonene Counteracts Brain Damage Induced by High Fat-Diet in Obese Mice"

_ijms, 2023, doi:10.3390/ijms24043467_

Round 1

Reviewer 1 Report

The Authors conclude that HFD causes detrimental effects on AD-related neuropathological and neuroinflammatory pathways leading to neurodegeneration. The long-term ingestion of Sicilian Black Bee Chestnut Honey, or D-limonene, separately or in combination, can counteract and  ameliorate the cerebral stressing conditions related to HFD-induced metabolic disorders. The HFD brains showed a higher number of apoptotic nuclei, gene upregulation of 22 Fas-L, Bim, P27, IL-1β, IL-6 and TNF-α and downregulation of BDNF and BCL2, higher expression of COX-2 and iNOS protein, higher levels of ROS and nitrite compared to STD. These alterations were improved in HFD-H, HFD-L and HFD-H+L, suggesting neuroprotection.

comments
- It would be important to evaluate the effect of honey and D-limonene in rats on a standard diet because neurodegenerative disorders in normalweight persons often have a genetic root related for example to insulin resistance, familial hypercholesterolemia, hypertension, cardiovascular disease
that often does not depend only on hyperlipidic diet intake.
-Explain how concentrations of Sicilian Black Bee Chestnut Honey and limonene administered to rats were determined
Discussion lines 207 208
The Authors report "Epidemiological human studies pointed out that a high calorie diet is associated with worse performance on a cognitive task"

Reference 31 reported an  associations between dietary intake and cognitive performance in elderly people aged 65-90 y who were free of significant cognitive impairment. Dietary intake was monitored with a weighed-food record for 7 consecutive days. this study evaluated elderly subjevcts  were more predisposed to age-related neurodegenerative changes and therefore diet has a more pronounced effect
-The authors report," In our experiments we used HFD obese mice, which following chronic consumption of HFD develop hyperglycemia, dyslipidemia, insulin resistance activation of amyloidogenic pathways, neuroinflammation and neurodegeneration; consequently, they are suitable for verifying the potential effects of functional food/phytochemicals on neuronal survival."
The mice were not obese but became obese with HFD. In fact, a reference group was treated with normo fat diet. This point is very important in  humans as usually obese patients have a genetic root associated with eating disorders.
-In conclusion, o results of the study are interesting. the authors should more focus on their results rather than hypothesize a positive effect of honey and limonene in the states of neurodegeneration, infismmazione and Alzheimer's that in humans are also evaluated with clinical evidence

Author Response

Comments of Review 1

Thank’s for your comments and suggestions.

  1. It would be important to evaluate the effect of honey and D-limonene in rats on a standard diet because neurodegenerative disorders in normal weight persons often have a genetic root related for example to insulin resistance, familial hypercholesterolemia, hypertension, cardiovascular disease that often does not depend only on hyperlipidic diet intake.    Reply: Thank for your suggestion. It is very interesting. In the next project we will sure include this groups of animals in order to analyze the protective effects against also potential development of MetS in predisposal lean individuals.
  2. Explain how concentrations of Sicilian Black Bee Chestnut Honey and limonene administered to rats were determined.                                     Reply: We have reported this information in the first version of manuscript (p.10, lines 318-320). In details, the honey’s dose (45 mg honey ingested/day/mouse) was extrapolated by the study of Nasuti, C. and coll. (Antioxidative and gastroprotective activities of anti-inflammatory formulations derived from chestnut honey in rats. Nutr. Res. 2006, 26, 130–137). The D-limonene dose (0.5% w/w) was obteined by the study of Jing, L.et al. Preventive and ameliorating effects of citrus D-limonene on dyslipidemia and hyperglycemia in mice with high-fat diet-induced obesity. Eur J Pharmacol 2013; 715, 46-55).
    3. Discussion lines 207 208. The Authors report "Epidemiological human studies pointed out that a high calorie diet is associated with worse performance on a cognitive task". Reference 31 reported an  associations between dietary intake and cognitive performance in elderly people aged 65-90 y who were free of significant cognitive impairment. Dietary intake was monitored with a weighed-food record for 7 consecutive days. this study evaluated elderly subjevcts were more predisposed to age-related neurodegenerative changes and therefore diet has a more pronounced effect.                                                                                                       Reply: Thank’s for your comments. We have used reference 31 with the aim to highlighted that, under the same conditions of seniority and potential age-related neurodegeneration, a diet rich of fat is strongly linked to the onset or worsening of cognitive decline, as indicated by the study of Ortega, R.                                                                                         4. The authors report," In our experiments we used HFD obese mice, which following chronic consumption of HFD develop hyperglycemia, dyslipidemia, insulin resistance activation of amyloidogenic pathways, neuroinflammation and neurodegeneration; consequently, they are suitable for verifying the potential effects of functional food/phytochemicals on neuronal survival."
    The mice were not obese but became obese with HFD. In fact, a reference group was treated with normo fat diet. This point is very important in humans as usually obese patients have a genetic root associated with eating disorders.

         Reply Thanks for your suggestion. We have changed the sentence.

5. In conclusion, o results of the study are interesting. the authors should more focus on their results rather than hypothesize a positive effect of honey and limonene in the states of neurodegeneration, infismmazione and Alzheimer's that in humans are also evaluated with clinical evidence.

Reply. Thanks for all the comments. We appreciate your point of view, but in our results we wanted to highlight just as a chronic intake of natural substances with high anti-inflammatory power could counteract neuronal damage that occurs in a high fat diet and then in an obesity state.  

Reviewer 2 Report

Thanks to submit "Long-Term Ingestion of Sicilian Black Bee Chestnut Honey and/or D-Limonene Counteracts Brain damage induced by High Fat-Diet in Obese Mice" to IJMS.

Overall the manuscript is interesting, well written and appears to have been conducted with scientific criteria. The research evaluated the potential of honey and/or limonene in the neurological damage caused by induced obesity. In fact, there are not many studies that focus on this topic. I felt a lack of pro/anti-inflammatory cytokine dosage in the blood. In my opinion the manuscript can be accepted after making minor corrections.

Line#17: only honey and D-limonene together?

Abstract: although I agree that the subject is interesting and at the same time complex, the abstract text is too dense for a common reader (who is not a specific researcher in the area). The use of abbreviations without definition is an example where the text needs to be modified. Perhaps instead of describing the results, they provide the reader with the interpretation of the results. Keeping the language simpler in the abstract can attract more readers to your manuscript.

Line#40: improve the spelling of this sentence, for example, I don't know if "they" is related to dementia and AD or obesity and diabetes.

Line#307: missing final point

Line#199: downregulated only. Why are some numbers positive and others negative then?

Line#189-193: Looks to me like these are the genes that were reported so I guess the best place to say how this table of results was generated is under material and methods. another issue here is to cite references that indicate these genes as important in this case. performing a scan to search for specific genes is interesting, but it is necessary to base oneself on what is described in the literature as important.

Check discussion item formatting.

Author Response

Comments of Review 2

  1. I felt a lack of pro/anti-inflammatory cytokine dosage in the blood. 1. Reply: Thanks for your comment. Indeed, we have performed dosages of plasma levels of TNFα and IL-1β, but technical problems came to invalidate the ELISA analysis; and not enough plasma was available to repeat the analysis. In another future study aimed at deepening the topic, we will perform a multiplex analysis to analyse all the variations occurring in the whole panel of inflammatory markers plasma concentration.

  1. Line#17: only honey and D-limonene together?2. Reply: Thanks for your question. We understood that it needs to clarify also in the abstract that we used honey and/or D-limonene, ingested separately or in combination. So, we have changed the sentence.
  2. Abstract: although I agree that the subject is interesting and at the same time complex, the abstract text is too dense for a common reader (who is not a specific researcher in the area). The use of abbreviations without definition is an example where the text needs to be modified. Perhaps instead of describing the results, they provide the reader with the interpretation of the results. Keeping the language simpler in the abstract can attract more readers to your manuscript. 3. Reply: According to your suggestion we modified the abstract in order to make it easier for readers of different scientific areas.
  3. Line#40: improve the spelling of this sentence, for example, I don't know if "they" is related to dementia and AD or obesity and diabetes. 4. Reply: Thanks for suggestion. We have modified the sentence.
  4. Line#307: missing final point. 5. Reply: Sorry, we have checked but not found where the point is missing.

  1. Line#199: downregulated only. Why are some numbers positive and others negative then? 6. Reply: The negative or positive number derives from the ratio of the gene expression levels between the two genes considered. So, when the result of the ratio is negative, indicates a lower expression of the target gene in the treated group (reported in the numerator, for example HFD-H group) compared to the expression of the same gene in the control group (reported in the denominator, for example HFD). In the same way, the positive numbers derive from the overexpressed gene in the numerator in comparison with the expression of the gene in the denominator.
  2. Line#189-193: Looks to me like these are the genes that were reported so I guess the best place to say how this table of results was generated is under material and methods. 7. reply: In truth, we used a PCR array that allows analysing in a contemporary manner the expression change of 96 genes involved in the onset and progression of AD. Among these 96 genes, we choose to analyze and describe only the genes showing a variation of gene expression more than the 2 fold-change among the different groups analyzed. To better describe this point we added sentences in the Material and Methods (p. 12; lines 418-419; 424-426)

Another issue here is to cite references that indicate these genes as important in this case. performing a scan to search for specific genes is interesting, but it is necessary to base oneself on what is described in the literature as important. Reply: Thanks for the comments. According to us, the opportune references were already added in the previous discussion. On the other way, genes reported in the plate of PCR array, are known markers of AD’s pathogenetic pathway.

8. Check discussion item formatting. Reply: Thanks. Done.
